# Kinetics and the Effect of Thermal Treatments on the Martensitic Transformation and Magnetic Properties in Ni$_{49}$Mn$_{32}$Ga$_{19}$ Ferromagnetic Shape Memory Ribbons

**Felicia Tolea** [ID]**, Bogdan Popescu** [ID]**, Cristina Bartha** [ID]**, Monica Enculescu** [ID]**, Mugurel Tolea** **and Mihaela Sofronie *** [ID]

National Institute of Materials Physics, Atomistilor 405A, 077125 Magurele, Romania
* Correspondence: mihsof@infim.ro

**Abstract:** In our work, the kinetics of martensitic transformations and the influence of thermal treatments on martensitic transformations, as well as the related magnetic properties of the Ni$_{49}$Mn$_{32}$Ga$_{19}$ ferromagnetic shape memory melt-spun ribbons, have been investigated. Thermal treatments at 673 K for 1, 4 and 8 h can be considered an instrument for fine-tuning the performance parameters of alloys. One-hour thermal treatments promote an improvement in the crystallinity of these otherwise highly textured ribbons, reducing internal defects and stress induced by the melt-spinning technique. Longer thermal treatments induce an important magnetization rise concomitantly with a slight and continuous increase in martensitic temperatures and transformation enthalpy. The activation energy, evaluated from differential scanning calorimeter experimental data with a Friedman model, significantly increases after thermal treatments as a result of the multi-phase coexistence and stabilization of the non-modulated martensitic phase, which increases the reverse martensitic transformation hindrance.

**Keywords:** kinetics; ferromagnetic shape memory alloys; melt-spun ribbons; martensitic transformations

## 1. Introduction

Ferromagnetic shape memory alloys (FSMAs) are a special class of smart materials extensively studied as important candidates in applications such as robotics [1,2], micropumps [3], biomedical devices [4], actuators [5,6], sensors [7], and other engineering applications [8–11]. The multi-functional properties of well-known Ni-Mn-Ga Heusler FSMAs are based on a martensitic transformation (MT), a thermo-elastic reversible structural phase transition in the magnetic state between a high-symmetry phase (austenite) and a lower one (martensite) [12]. On cooling, the disordered B2 or ordered L2$_1$ structure of austenite undergoes a diffusionless transformation with atoms shifting cooperatively to a low-symmetry modulated (five-layer (5M) and/or seven-layer (7M)) or non-modulated (tetragonal L1$_0$) martensite structure, depending on the composition, valence electron concentration (e/a), and thermal history of the alloy [13–16]. Inter-martensite transformations (IMTs) are not uncommon, and they are determined by the tendency of the tetragonal NM phase to be stable, which forms to the detriment of the metastable modulated 5M and 7M [17]. The stoichiometric Ni$_2$MnGa Heusler FSMA exhibits MT at 202 K and a ferromagnetic order below 376 K [13], while the off-stoichiometric alloys show a large variation in MT temperatures, up to the Curie temperature (T$_C$), depending on composition [14].

Ni$_2$MnGa-type alloys are produced in various forms, such as single crystals, bulk, ribbons, and thin films [15]. The fabrication of single crystalline samples requires high costs and laborious procedures, followed by prolonged annealing stages; for this reason, they are unfeasible for applications. The bulks are also difficult to process because of their intrinsic brittleness; moreover, undesirable phase precipitation or decomposition may occur, which alter the functional properties of the alloys [18]. Employing the melt-spinning

technique, textured ribbons with a non-equilibrium structure are obtained readily in a single-step process, at an almost-ready shape for engineering applications, and with a microstructure that favors improved elasticity. The melt-spun ribbons have lower MT and Curie temperatures than single crystals or bulk alloys, with a wide temperature range that is tunable by changing the melt-spinning speed or valence electron concentration (e/a) ratio [19–23]. Various thermal treatment schemes induce structural and magnetic property changes and may be considered an additional factor of property adjustment to meet specific application criteria. The effect of thermal treatments on the MT temperature and $T_C$ at high temperatures (~1000 K) is well covered by numerous studies [24–27]. During MT, the properties of Ni-Mn-Ga FSMAs are affected by heating rates, which suggests that by controlling the heating speed [28], the performance parameters of the alloys can be changed conveniently. The kinetics of the MT are dominated by the nucleation processes in avalanches [29] and the collective movement of a rather large number of atoms [30].

In this work, the effect of different thermal treatments at a rather low temperature on the microstructure, the MT kinetics, and the related magnetic properties of the polycrystalline Ni-Mn-Ga ribbons are studied. The valence electron concentration (e/a = 7.71) ratio indicates that the martensitic transformation is above room temperature, and close to the Curie temperature for the $Ni_{49}Mn_{32}Ga_{19}$ alloy, with important implications in its multifunctionality. The characteristics of the MT and the thermal behavior of the $Ni_{49}Mn_{32}Ga_{19}$ as-prepared and thermally treated ribbons are discussed, and the kinetic parameters, especially the activation energies, are obtained by employing a non-isothermal model—the Friedman model. The microstructure, investigated by scanning electron microscopy (SEM) and temperature-dependent X-ray diffraction (XRD), is discussed in correlation with magnetic analyses.

## 2. Materials and Methods

Polycrystalline ingots with a $Ni_{49}Mn_{32}Ga_{19}$ nominal composition were obtained by electric-arc melting high-purity elements Ni (99.99%), Mn (99.95%), and Ga (99.99%) under an argon atmosphere. Bulk alloys were flipped and remelted four times to ensure their homogeneity. Subsequently, the ingots were induction melted under an argon protective atmosphere in a quartz crucible with a circular nozzle (0.5 mm diameter). Ribbon-shaped specimens were obtained after the melt was ejected by applying an argon overpressure flux (40 kPa on the polished water-cooled Cu wheel, rotating at a constant speed (linear velocity of 20 m/s). These thin as-prepared ribbons (denoted Ga-0h) with a width of about 2–3 mm and a thickness of about 13–15 μm, were thermally treated in vacuum quartz ampoules at 673 K for 1 h, 4 h, and 8 h, and then rapidly quenched in ice-cooled water to promote the martensitic phase in the samples. The thermally treated samples were denoted Ga-1h, Ga-4h, and Ga-8h, respectively.

The martensitic transformation and its characteristic parameters were studied using thermal analysis measurements. Cooling and heating cycles with a 20 K/min scanning rate, under a He atmosphere, were carried out in the 300–400 K temperature range with a differential scanning calorimeter (DSC) model 204 F1 Phoenix (Netzsch). The accuracy of the heat-flow measurements was $\pm$ 0.001 mW, and the temperature precision was $\pm$0.01 °C. Furthermore, the DSC measurements at different scanning rates (5, 10, 12, 15, and 20 K/min, respectively) were achieved to evaluate the MT kinetic parameters and to obtain the activation energy (E).

Any kinetic analysis based on free models yields the activation energy and the preexponential factor of a process/reaction without assuming a kinetic model [31]. For non-isothermal conditions the following equation is used:

$$\frac{d\alpha}{dt} = \frac{A}{\beta} \exp\left(-\frac{E}{RT}\right) f(\alpha) \tag{1}$$

where $\alpha$ is the conversion, T is the temperature, $\beta$ is the heating rate, $f(\alpha)$ is the reaction model, R is the universal gas constant, and E and A are the kinetic parameters (activa-

tion energy and frequency factor), respectively. The mathematical description of any process/reaction is made with the kinetic triplet (i.e., E, A and $f(x)$).

In non-isothermal conditions, the analytical integration of Equation (1) can be expressed as follows:

$$g(\alpha) = \frac{A}{\beta} \int_0^T \exp(-\frac{E}{RT})dT \qquad (2)$$

Equation (2) does not have an analytical solution and can be solved only by the numerical integration of approximations [32]. The simplest approximations are given by the equations corresponding to the free models: Friedman, Ozawa–Flynn–Wall, Kissinger, etc. [33–35]. The Friedman model that is used in this paper is based on an inter-comparison of the conversion rate ($\frac{d\alpha}{dt}$) for a particular degree of conversion ($\alpha$) determined at different heating rates [33]. This method is described by the following logarithmic differential equation:

$$log\frac{d\alpha}{dt} = log\alpha\frac{d\alpha}{dt} - logAf(\alpha) - \frac{E}{4.575T} \qquad (3)$$

With this model, both the activation energy (E) and pre-exponential factor (A) are determined by plotting $log\frac{d\alpha}{dt}$ versus $\frac{1}{T}$ for a constant $\alpha$ value.

Structural investigations at different temperatures (300–370 K) were collected using a Rigaku-SmartLab X-ray diffractometer (Rigaku Corporation, Tokyo, Japan) with Cu radiation (Cu = 1.5406 Å) in Bragg–Brentano geometry equipped with an additional DHS 1100 temperature chamber (Anton Paar GmbH, Graz, Austria). The XRD measurements on the contact side with the copper wheel (CS) and the free side (FS) prove the expected behavior of the highly textured ribbons (not shown). Ribbons were glued on a copper holder using a silver paste to ensure thermal contact. Therefore, the X-ray patterns show the reflections characteristic of cubic (F$m$-$3m$) Cu and Ag as a consequence of the experimental setup. The XRD patterns were indexed using Bruker AXS DIFFRAC. EVA software (Bruker AXS, Karlsruhe, Germany, 2000). The data were analyzed by the Le Bail method, employing FullProf Suite software.

The surface and cross-section morphology, as well as the chemical composition, of the as-prepared and thermal-treated samples were examined by Scanning Electron Microscopy (SEM) and Energy Dispersive X-Ray Spectroscopy (EDS), employing a Zeiss Evo 50 XVP microscope equipped with a Bruker EDS detector at RT. All samples were cleaned with a HF(5%)-HNO$_3$(5%)-H$_2$O solution before analysis. The magnetic measurements were performed by a Quantum Design superconducting quantum interference device (SQUID) (San Diego, CA, USA) in the Reciprocal Space Option (RSO) mode and in the temperature range of 300–400 K, with the magnetic field (up to 4 T) applied along the ribbon length.

### 3. Results

*3.1. XRD*

The structural evolution of the Ga-8h sample with temperature is shown in Figure 1 and is representative of all samples. At room temperature (RT), the highly textured ribbons are characterized by the coexistence of two martensite crystalline phases: 7M monoclinic, belonging to the *I2/m* space group; and non-modulated (NM) tetragonal with the *I4/mmm* space group, alongside the austenitic L2$_1$ cubic structure (*Fm-3m*). The patterns and data analyses show that with increasing temperature, the martensitic phases gradually transformed into the austenitic cubic L2$_1$ phase. The existence of the two martensitic phases indicates a possible inter-martensitic transformation (7M–NM) [36]. Similar findings were reported in alloys with close compositions and with transformation temperatures in the 325–353 K region [37].

The lattice parameters at RT of all samples are given in Table 1. Their evolution with temperature is summarized in Figure 2, together with that of unit cell volume. It can be immediately observed that the thermal treatments produced an increase in the temperature

at which the three phases coexist, from approximately 330 K in the Ga-0h sample to around 343 K in the Ga-8h sample.

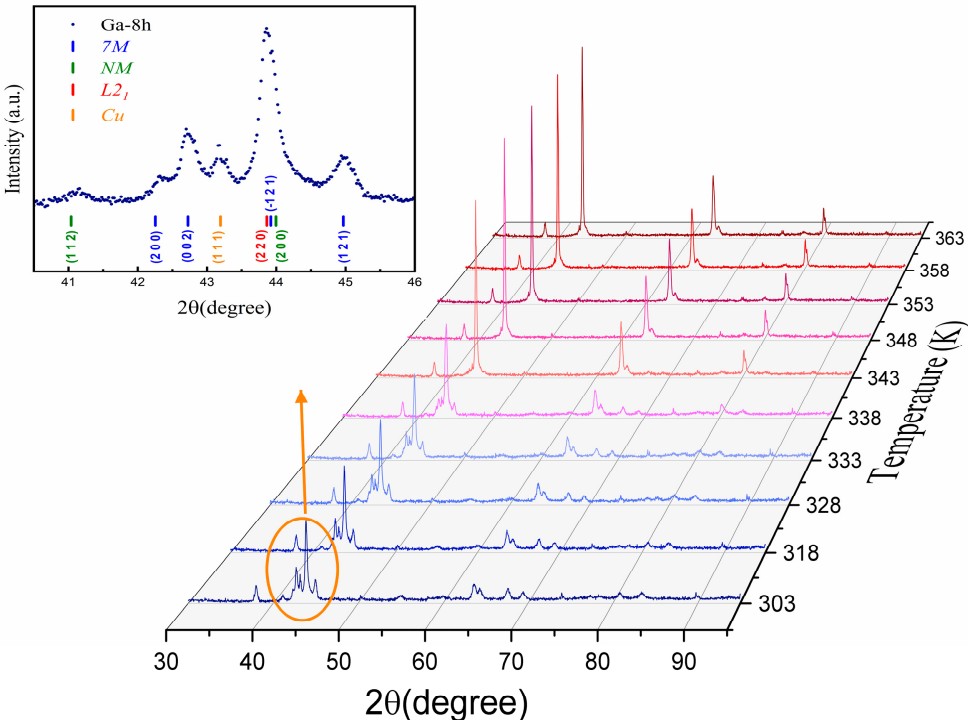

**Figure 1.** The evolution of the XRD patterns with temperature for the Ga-8h sample.

**Table 1.** The lattice parameters at room temperature.

| Sample | Phase | a (Å) | b (Å) | c (Å) | β (Degree) | V (Å³) | c/a |
|--------|-------|-------|-------|-------|------------|--------|-----|
| **Ga-0h** | 7M (*I2/m*) | 4.276(3) | 5.445(7) | 4.234(1) | 90.541 | 98.597(8) | 0.990(1) |
| | NM (*I4/mmm*) | 4.094(5) | - | 6.730(6) | - | 112.838(4) | 1.643(8) |
| | L2$_1$ (*Fm-3m*) | 5.839(1) | - | - | - | 199.085(8) | - |
| **Ga-1h** | 7M (*I2/m*) | 4.267(5) | 5.447(2) | 4.227(9) | 90.536 | 98.276(8) | 0.990(7) |
| | NM (*I4/mmm*) | 4.075(6) | - | 6.737(9) | - | 111.918(3) | 1.653(2) |
| | L2$_1$ (*Fm-3m*) | 5.833(4) | - | - | - | 198.497(5) | - |
| **Ga-4h** | 7M (*I2/m*) | 4.273(4) | 5.438(4) | 4.228(6) | 90.489 | 98.272(7) | 0.989(5) |
| | NM (*I4/mmm*) | 4.098(1) | - | 6.717(6) | - | 112.820 | 1.639(1) |
| | L2$_1$ (*Fm-3m*) | 5.829(6) | - | - | - | 198.118(2) | - |
| **Ga-8h** | 7M (*I2/m*) | 4.271(4) | 5.437 | 4.228(9) | 90.501 | 98.206(3) | 0.99 |
| | NM (*I4/mmm*) | 4.112(9) | - | 6.710(2) | - | 113.509(7) | 1.631(5) |
| | L2$_1$ (*Fm-3m*) | 5.833(1) | - | - | - | 198.475(6) | - |

The c/a ratio is considered one of the factors that influence transformation temperatures, with lower values corresponding to low transformation temperatures [38]. Because only the modulated martensites have c/a < 1, it can be inferred that the presence of the 7M phase determines the reduced value of the transformation temperatures, around RT and below T$_C$. Around 330 K, a sudden variation in the unit cell volume for the 7M and NM phases could be the signature of the inter-martensitic transformation (Figure 2b). Figure 2a shows that, with increasing temperature, the lattice parameter (*a*) of the L2$_1$ phase increased, and the volume expansion of the unit cell was observed, taking maximum values after the transformation is completed (Figure 2b).

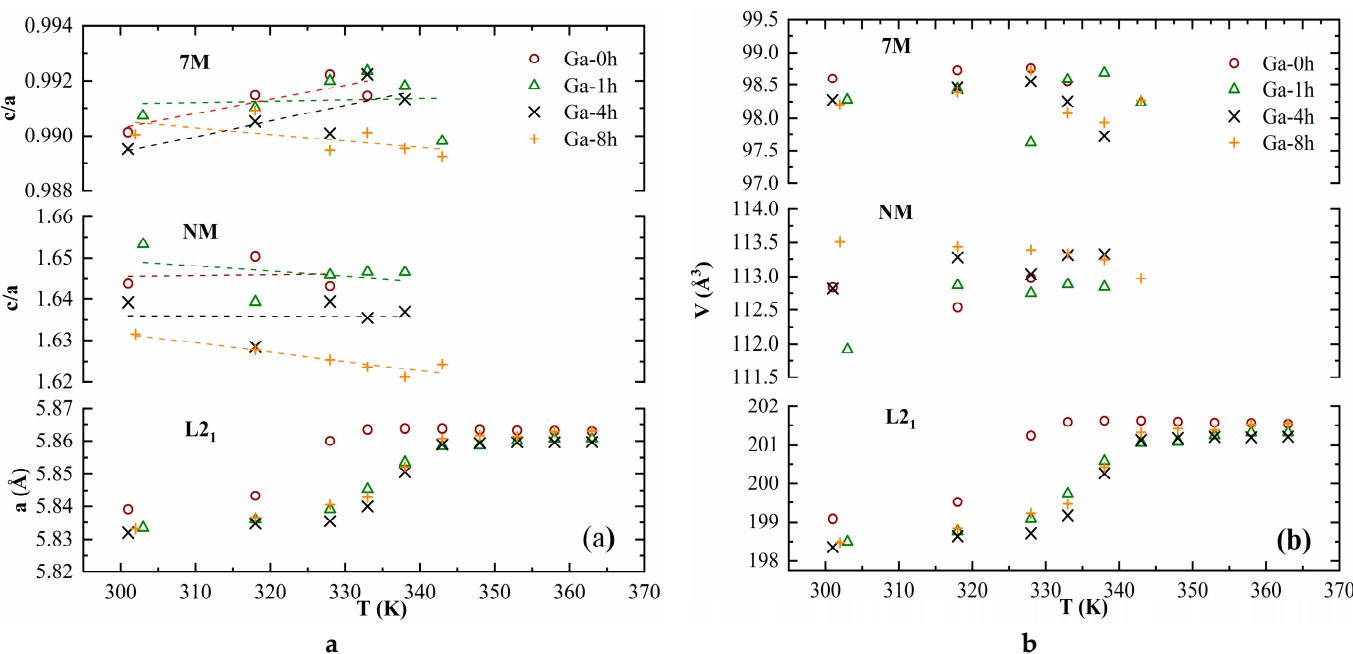

**Figure 2.** The evolution with the temperature of the lattice parameters (**a**) and unit cell volume (**b**).

### 3.2. DSC and Kinetics

The cooling and heating DSC scans revealed exothermal and endothermal peaks, which are a signature of thermoelastic and reversible martensitic transformations (Figure 3). MT occurs at temperatures above RT, increasing slightly with the duration of the sample thermal treatments. On the cooling and heating curves, at temperatures higher than MT, a small peak was observed, which indicates the Curie temperature ($T_C$) and confirms the ferromagnetic behavior of the samples. The specific MT temperatures (martensite start -Ms and martensite finish -Mf; austenite start -As and austenite finish -Af) and Curie temperatures were determined on DSC curves using the tangential line method (shown with black arrows in Figure 3), listed in Table 2. The transformation enthalpy, H, associated with the DSC peak area, was calculated as the average between the forward and reverse transformation enthalpy. The thermodynamic equilibrium temperature $T_0$, defined by the equality of Gibbs energy of the martensite and austenite, was calculated by the relation $T_0 = (Ms + Af)/2$ [39].

**Table 2.** The MT characteristic parameters: temperatures (martensite start (Ms) and martensite finish (Mf); austenite start (As) and austenite finish (Af)); the thermodynamic equilibrium temperature ($T_0$); and transformation enthalpy (H) for as-prepared and thermal-treated samples. The Curie temperatures were obtained from DSC ($T_{C-DSC}$) and magnetic measurements($T_{CA}$ and $T_{CM}$) for austenite and martensite, respectively (were extracted from extrapolation of Arrott curves).

| Sample | Ms // Mf (K) | As // Af (K) | H (J/g) | $T_0$ (K) | $T_{C-DSC}$ (K) | $T_{CM}$ // $T_{CA}$ (K) |
|--------|--------------|--------------|---------|-----------|-----------------|--------------------------|
| **Ga-0h** | 336 // 325 | 334 // 343 | 6.205 | 339.5 | 357.5 | 348 // 345 |
| **Ga-1h** | 341 // 329 | 339 // 348 | 6.84 | 344.5 | 359 | 400 // 362 |
| **Ga-4h** | 340 // 329 | 338 // 347 | 6.6 | 343.5 | 359.5 | 380 //356 |
| **Ga-8h** | 344 // 330 | 340 // 351 | 7.27 | 347.5 | 360.5 | 382 //357 |

The DSC measurement analysis indicates that MT temperatures increased slightly and continuously (~8 K) beside transformation enthalpies (up to 7.27 J/g for Ga-8h) with an almost constant Tc value, which imposed low-temperature thermal treatments as an instrument for fine-tuning them. The Ga-4h and Ga-1h parameters' behavior was similar, which implies that four hours of thermal treatment is not enough to produce noticeable structural changes. The range of martensitic transformation (Af − Mf~18 K) and thermal

hysteresis (Af − Ms~7 K) remained constant, and the transformation moved towards higher temperatures.

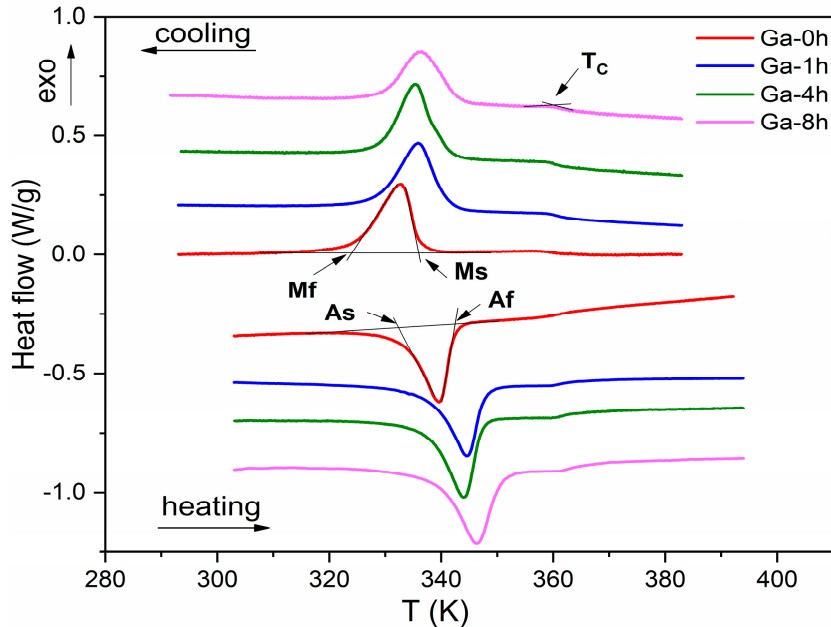

**Figure 3.** DSC scans on the cooling and heating cycles for as-prepared (Ga-0h) and thermally treated samples (Ga-1h, Ga-4h, and Ga-8h, respectively).

A non-isothermal kinetic model, the Friedman model, was used to evaluate the activation energy (E) and to understand the effect of thermal treatments on the martensitic transformation for as-prepared and thermal-treated samples. For this reason, the DSC measurements at different heating rates were carried out for all samples (Figure 4a illustrates a typical example for the Ga-0h sample). With this model, both the activation energy (E) and the pre-exponential factor (A) were determined by plotting $log\frac{d\alpha}{dt}$ versus $\frac{1}{T}$ for a constant partial area $\alpha$ value (Figure 4b for Ga-0h sample). Figure 4c–e shows the dependence of the kinetic parameters on the partial area ($\alpha$) of the transformed austenite from the martensite during MT, for all ribbons. The DSC curves at different heating rates for the Ga-1h, Ga-4h, and Ga-8h samples, and the 1/T dependence of log d$\alpha$/dt for thermally treated ribbons are supplied as supplementary materials (Figures S1a–c and S2a–c).

The model-free analysis results for the as-prepared ribbons (Ga-0h) reveal that both kinetic parameters had nonlinear behavior as a result of the atomic disorder and internal stress stored in the ribbons after the melt-spinning processing. The activation energy curve started at 133 kJ/mol and had two broad peaks at partial area $\alpha$ = 0.3 (E = 145 kJ/mol) and $\alpha$ = 0.75 (E = 140 kJ/mol), respectively (Figure 4c). The unusual increase observed after $\alpha$ = 0.95 might be an effect of the high scanning rate at 20 K/min. The degree of order or internal stress is important for austenite nucleation and influences the path of reversible martensitic transformation [29].

The one-hour thermal treatment generated almost linear behavior in the kinetic parameters, with a two-fold increase in the start activation energy, compared to the as-prepared ribbons (E = 280 kJ/mol), and a rapid decrease above $\alpha$ =0.7 (E = 170 kJ/mol) (Figure 4d). After four hours of thermal treatment, a higher activation energy (E = 530 kJ/mol) is required to start the reverse martensitic transformation, although this suddenly decreased at partial area $\alpha$ = 0.15 (E = 180 kJ/mol) and then raised slightly again up to E = 240 kJ/mol at $\alpha$~0.75 (Figure 4e). Finally, the last thermal treatment for 8 h at 673 K induced a significant increase in the start activation energy (E = 725 kJ/mol), five-fold that of the Ga-0h sample (Figure 4f). In this case, the kinetic parameter curves have a continuous and rapid decrease until the martensitic transformation is complete.

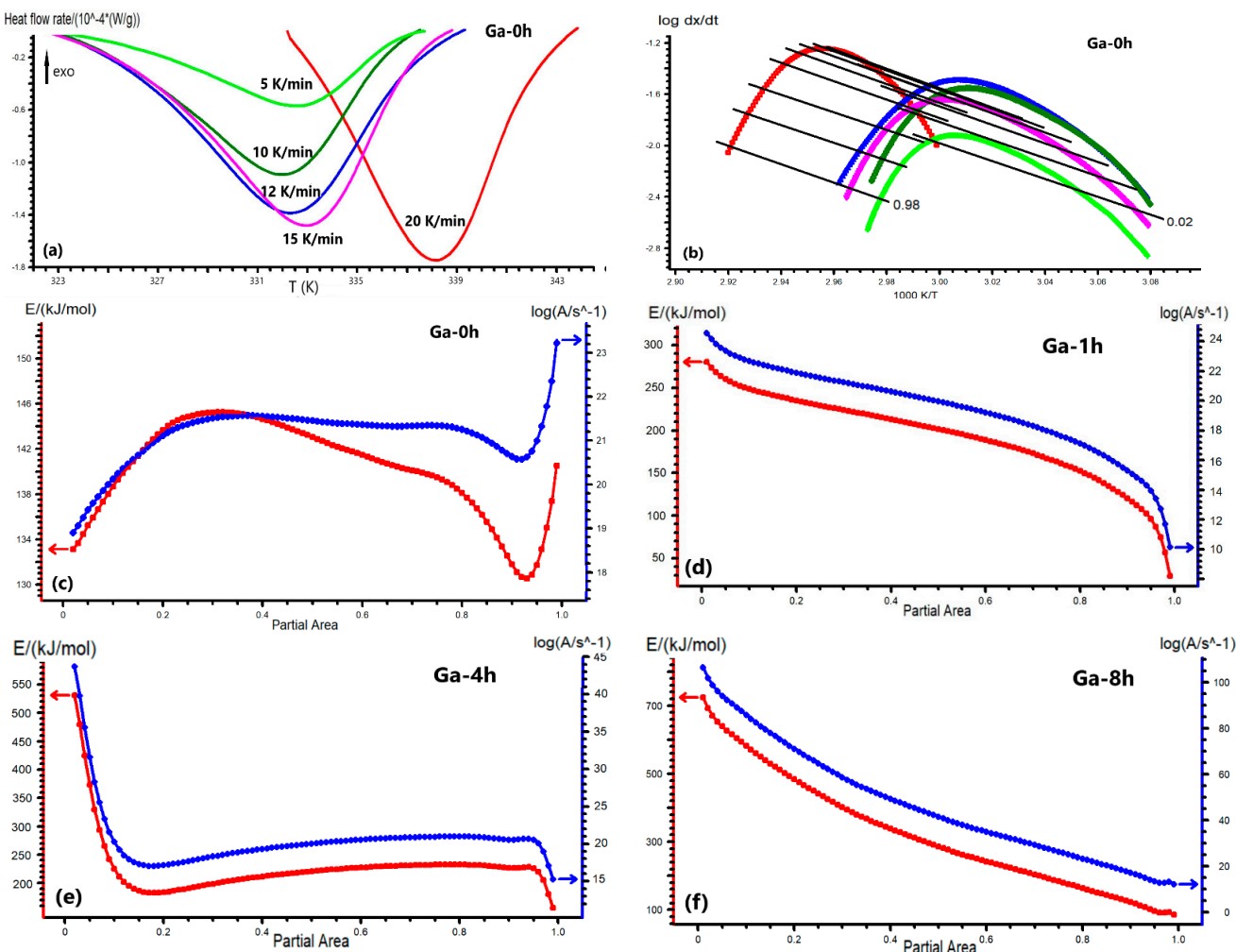

**Figure 4.** DSC curves at the different heating rates (5, 10, 12, 15, 20 K/min) for the Ga-0h sample (**a**); the 1/T dependence of log d$\alpha$/dt according to the Friedman model for the Ga-0h sample (**b**); the dependencies of the activation energy and pre-exponential factor by the transformed fraction ($\alpha$) according to the Friedman model for the Ga-0h (**c**), Ga-1h (**d**), Ga-4h (**e**), and Ga-8h (**f**) samples, respectively.

As previously mentioned, thermal treatments release internal stress and reduce the crystal defects of the ribbons, which induce a slight increase in the size of the grains (see the next section). J. Wang et al. [40] found that grain size mostly influences the kinetics of the forward martensitic transformation, less so that of the reverse transformation. The coexistence at room temperature of the different phases (7M-monoclinic, NM-tetragonal, and L$2_1$-cubic) enforces additional barriers and activation energies for different stages of the reverse martensitic transformation. These multiple obstacles need a much larger driving force and a higher activation energy. A continuous increase in the activation energy value with thermal treatments is due to the stabilization of the NM martensite at room temperature, as indicated by X-ray diffraction (Figure 2b and Table 1). Z.Li et al. [36] reported that the 7M–NM inter-martensitic transformation brings extra transformation barriers to the reversible MT due to their crystal lattice distortion and the interfacial energy change of their plate interfaces. High values for activation energy were reported for the Ni$_{47.92}$Mn$_{37.5}$In$_{12.5}$Co$_{2.08}$ [28], Ni$_{50}$Mn$_{35}$In$_{15}$ [41], and Ni$_{55}$Fe$_{19}$Ga$_{26}$ ribbons [42].

*3.3. SEM*

Scanning electron microscopy (SEM) was used to analyze all samples' surface and cross-section morphologies at room temperature (Figure 5). In Figure 5a, the contact surface (CS) with the copper wheel of the as-prepared ribbons (Ga-0h) shows dendritic

(2–4.5 μm) and cellular (0.4–1.9 μm) grains clustered in colonies, without visible cracks or any precipitates inside and at their boundaries. On the fractured cross-section, the misoriented columnar grains can be seen with different lengths and thicknesses, which cannot span the entire ribbon thickness from the contact surface (CS) with the copper wheel to the free surface (FS) (Inset Figure 5a). The columnar microstructure promoted the highly textured behavior of the ribbon samples. The high-temperature gradient during the melt-spinning technique induced grain refinement and a dendritic structure in the off-stoichiometric Ni-rich Heusler alloys. Additionally, it stimulated the fast nucleation and the grain growth process along the cross-section during the melt's rapid cooling [43].

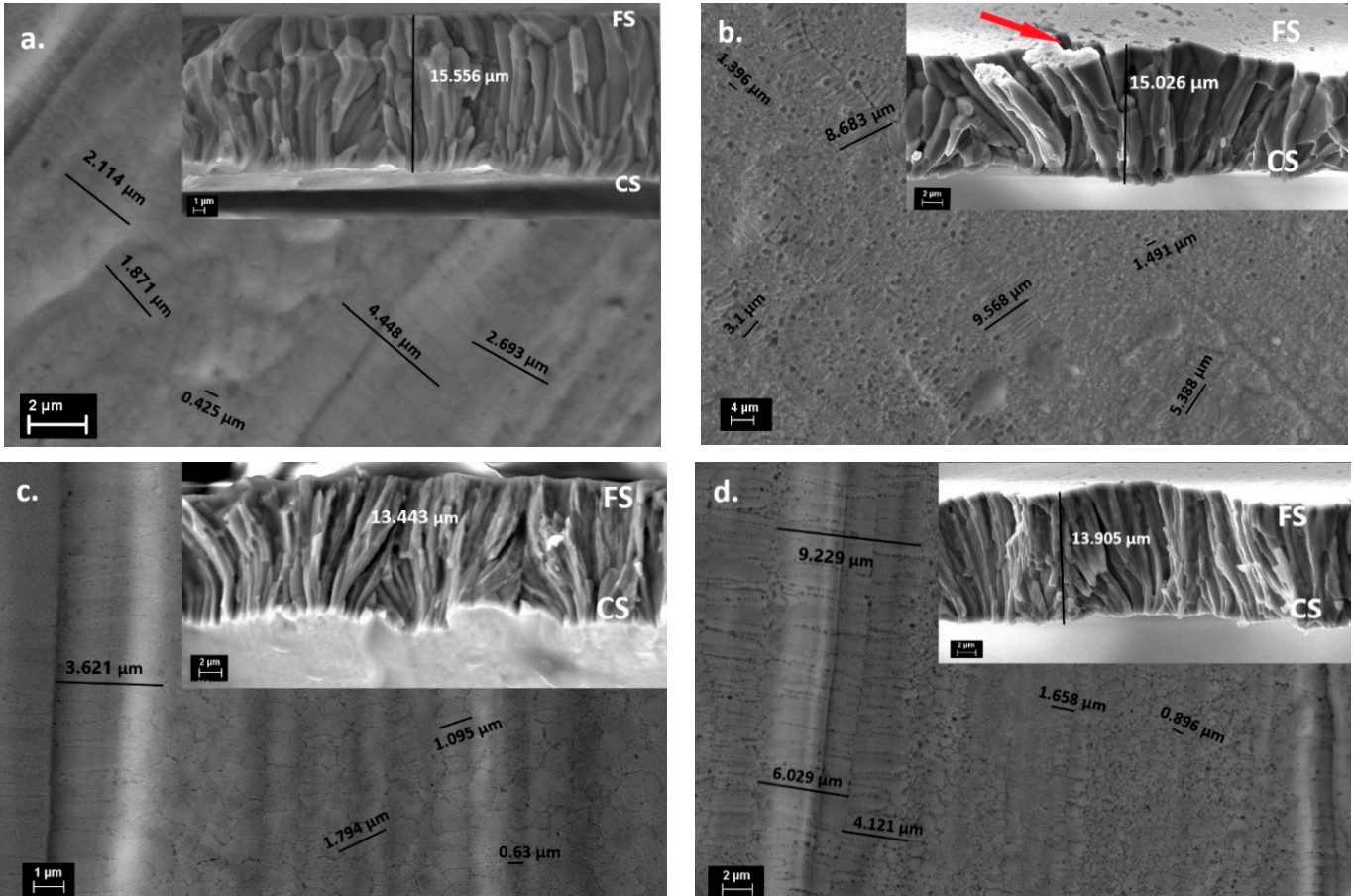

**Figure 5.** The contact surface SEM images (insets with cross-section images) for Ga-0h (**a**), Ga-1h (**b**), Ga-4h (**c**), and Ga-8h (**d**).

The one-hour thermal treatment at 673 K promoted the increase in dendritic grain size up to 9.5 μm and up to ~1μm in the smaller cellular ones due to stress release (Figure 5b). Moreover, cracks appeared on the free surface (FS) when the columnar structure spanned the cross-section of the ribbons (shown with red arrows in Figure 5b). The holes on the ribbon's surface were the result of the corrosive liquid action used for cleaning. The thermal treatment for 4 h at 673 K induced an increase, especially in the cellular grains size (0.6–1.8 μm) (Figure 5c), and a refinement of the misoriented columnar structure that spanned the entire cross-section with no evident cracks (shown in Figure 5c). Finally, after the last thermal treatment for 8 h at 673 K, the surface morphology of the Ga-8h ribbons showed both a dendritic and cellular grain size increase similar to the Ga-1h ribbons as a result of the defects' density reduction (Figure 5d), a well-oriented fine columnar structure, and no visible cracks on the cross-section image (shown in Figure 5d).

The smallest grains clustered in colonies and grew slowly after thermal treatments at a low temperature, while some of them retained the austenitic L2$_1$ cubic phase. D. Rajkumar et al. [44] claimed that the martensitic microstructure nucleates and develops in larger grains first, while in smaller ones, nucleation mainly starts in high-energy regions. Martensitic transformation temperatures are dependent on the grain size and increase with them. As such, the dendritic and cellular grains' predisposition to increasing in or maintaining size during the thermal treatment period induced the martensitic transformation temperature variation for our samples. It is important to mention that the highest and uniform grain size was reported only after higher-temperature thermal treatments [20]. The EDS analysis indicates that the chemical composition was the nominal one (Ni$_{49}$Mn$_{32}$Ga$_{19}$) for the as-prepared and thermally treated samples within the limits of the method's accuracy (~1.5 at%).

### 3.4. Magnetic Properties

A calculation of the Curie temperature (Tc) from the first derivative of magnetization, with respect to the temperature, is used frequently, but it is somehow insubstantial [45] and enables an estimation for the austenite phase only. Alternately, the Arrott plot method [46], based on the Ginsburg–Landau mean field frame/theory for magnetism, allows us to evaluate the Curie temperature for both the austenitic and martensitic phases. Therefore, the magnetization isotherms were measured up to a maximum of 4 T at different temperatures along the martensitic transformation (300–351 K), with a temperature variation of ~5 K. Figure 6a presents these measurements for the Ga-4h ribbons. The Arrott representation for the Ga-4h sample obtained from the magnetization isotherms is shown in Figure 6b.

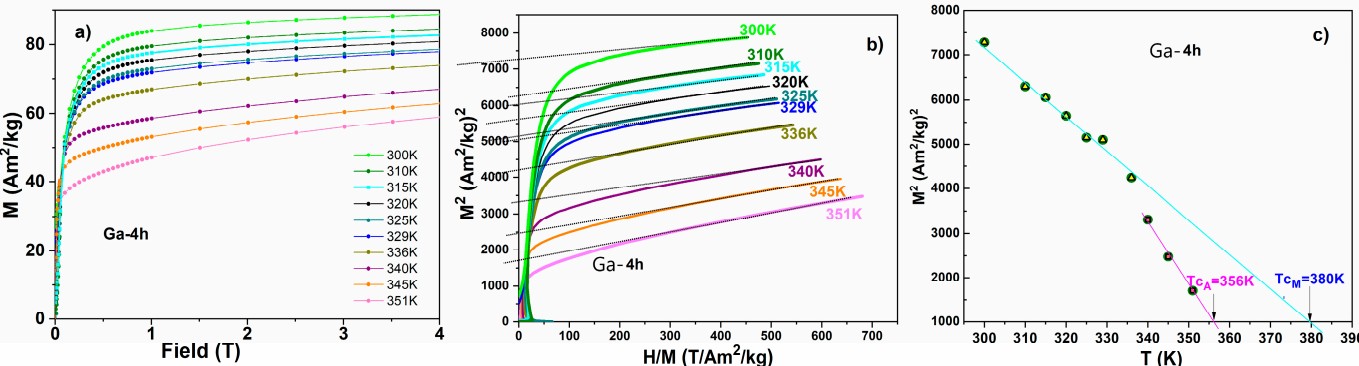

**Figure 6.** Magnetization isotherms measured up to 4 T at different temperatures along the martensitic transformation for Ga-4h sample (**a**); Arrott plot and spontaneous magnetization Ms obtained from linear extrapolation for Ga-4h sample (**b**); temperature dependence of spontaneous magnetization obtained from extrapolation of Arrott curves for Ga-4h sample (**c**). Curie temperatures of austenite and martensite were determined by extrapolating zero magnetization.

From M$^2$ versus H/M dependence, the square of spontaneous magnetization at zero field Ms$^2$ was determined by extrapolation [47]. The Curie temperatures of the austenite and martensite were then determined from the linear extrapolation of Ms$^2$ versus T to zero magnetization, as shown in Figure 6c. Accordingly, the values of magnetic ordering temperatures (Tc$_A$ for austenite and Tc$_M$ for martensite) were obtained (the values are given in Table 2), proving ferromagnetic interactions in both phases. The difference between Tc$_A$ and Tc$_M$ can be correlated with the increase in magnetization saturation during the martensitic transformation and suggest increased exchange interactions in martensite. Likely, these are indirect exchange magnetic interactions that depend on interatomic distances [48]. As seen in the data presented in Table 2, the Curie temperatures of austenite and martensite for the studied ribbons vary with the annealing time. Thus, for the Ga-0h ribbons, the difference between Tc$_A$ and Tc$_M$ was 3 K. The thermal treatment of 1 h to 673 K induced structural relaxation and the release of the tensions that are inherent

after the melt-spinning technique. Additionally, the difference between $Tc_A$ and $Tc_M$ became 38 K. After 4 h, thermal treatment $Tc_M$ was 24 K larger than $Tc_A$, and after the 8-h thermal treatment, martensite had a higher $Tc_M$, at 25 K, than $Tc_A$.

The thermo-magnetic measurements (Figure 7) performed on all studied ribbons revealed magnetic properties, alongside the martensitic transformation and magnetic ordering transition. As discussed in our previous works [49,50], the MT, as a first-order phase transition, influences the FSMAs' magnetic properties. The common signature is thermal hysteresis in magnetization around the MT temperature. The magnitude of the hysteresis depends not only on the values of the characteristic MT temperatures (Ms, Mf, As, Af), but also on the magnetic properties of martensite and austenite. Thermal treatments strongly influence the crystalline structure and the magnetic properties; therefore, thermal treatment at 673 K may initiate the atoms' and vacancies' diffusion, resulting in crystallization in the highly ordered $L2_1$ structure, as evidenced by XRD patterns. According to [51], atomic ordering processes take place following atomic diffusion and are mediated by vacancies. At the same time, the effect of the annealing time is also very important. Thus, the sample annealed for 1 h at 673 K showed a slight increase in Curie temperature and a higher magnetization compared to the Ga-0h sample, proving stronger magnetic interaction. As the thermal treatment time increased, the magnetization of the martensite increased (see Figure 7). At the same time, between the M (T) curves recorded for cooling and heating, there is a hysteresis that cannot be associated with martensitic transformation, because the maximum value for Af is 351 K. This is due to magnetic fluctuations around the Curie temperature and the differences between the $Tc_A$ and $Tc_M$ Curie temperatures, differences that are highlighted by the Arrott plot method (see Table 2).

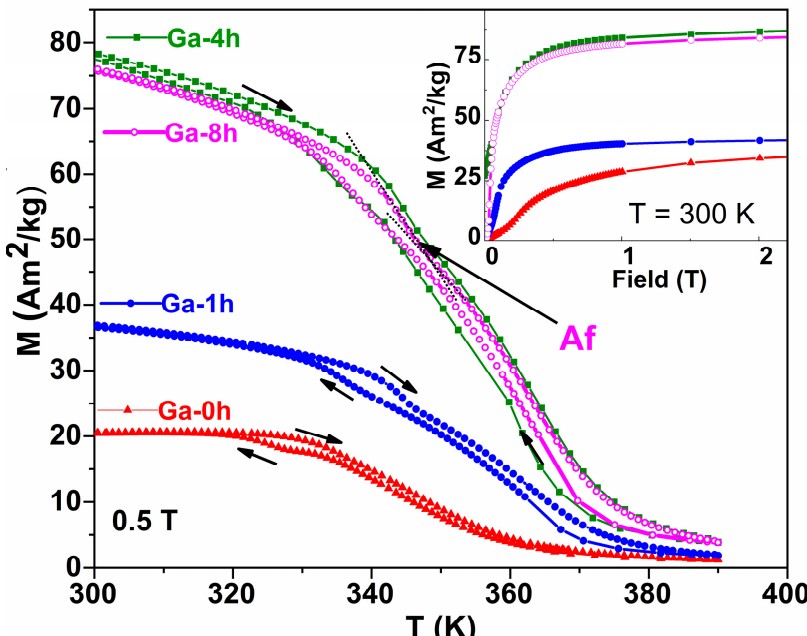

**Figure 7.** Thermomagnetic measurements were performed in 0.5 T applied magnetic field for investigated ribbons. Inset: magnetization curves recorded at 300 K.

Importantly, Figure 7 shows that at high temperatures, the magnetization of thermally treated samples does not decrease to zero. This fact might indicate the presence of another magnetic phase with a Curie temperature above 390 K. This is consistent with the segregation of the $\gamma$ secondary phase with a face-centered cubic structure, not participating in the thermoelastic martensitic transformation, but which has an important effect on the MT and the reorientation of martensitic variants through its size, shape, and distribution in the austenitic matrix, as reported in the literature [52,53]. Although this phase was not detected in the XRD patterns (being under the detection limit of the diffractometer of ~5%) or on the

SEM images, the temperature dependence of the magnetization in Figure 7 reveals another effect of the thermal treatments performed in the studied ribbons, namely the possible segregation of the secondary $\gamma$ phase which has Tc greater than 390 K. The $\gamma$ phase also contributes to the high magnetization of the Ga-4h and Ga-8h samples because this depletes the austenitic matrix in Ni (3d element) [54].

The inset of Figure 7, which shows the magnetization curves recorded at 300 K, highlights the increase in the saturation magnetization of martensite as the annealing time increases. This increase in magnetization (by 2.4 times for sample Ga-4h) is accompanied by an increase in Tc and the magnetization difference between austenite and martensite observed during MT (from the temperature dependence of magnetization measured in 0.5 T applied magnetic field) as a result of a larger magnetocrystalline anisotropy of the martensitic phase [55]. According to thermal analysis and XRD results, a thermal treatment time of 4h is not enough to produce noticeable structural changes (the Ga-4h and Ga-1h parameters' behavior is similar). Bearing in mind that the $\gamma$ phase (which also contributes to the high magnetization of the Ga-4h and Ga-8h samples) is present in a small amount and undetectable by XRD, another mechanism could explain the significant increase in magnetization after long-term treatments. A very recent study [56] quantitatively discusses vacancy-mediated diffusion as the main atomic mechanism responsible for the ordering process and changes in Ni-Mn-Ga Heusler alloy properties (Ms, $T_C$, or MT). The same authors, by comparing the calculated positron lifetime values associated with Ni vacancies [57] and the experimental values obtained from quenched Ni-Mn-Ga alloys [58,59], conclude that the ordering process is assisted by Ni vacancies, with the migration energy of Ni vacancies being one of the key parameters which govern the atomic ordering process in Ni-Mn-Ga alloys.

## 4. Conclusions

Polycrystalline ribbons with a $Ni_{49}Mn_{32}Ga_{19}$ nominal composition, prepared by the melt-spinning technique and thermally treated for different periods (1 h, 4 h, and 8 h) at a low temperature (673 K), have been studied. The high-textured as-prepared ribbons have a columnar microstructure in the cross-section and dendritic and cellular grains clustered in colonies, without visible precipitates on the surface. The structure of the as-prepared ribbons has evidenced the coexistence of the martensite structure with 7M monoclinic and non-modulated tetragonal phases and austenite with an $L2_1$ cubic structure at room temperature. Thermal treatments release internal stress and reduce the crystal defects of the ribbons, which induce a small increase in the grain size. The martensitic transformation temperatures increase slightly and continuously (~8 K), together with the transformation enthalpy. The substantial increase in activation energy (up to 750 kJ/mol) evaluated by the Friedman non-isothermal kinetic model sustains the shift in the martensitic transformation to higher temperatures as a result of the stabilization of the non-modulated martensite phase. At room temperature, the coexistence of the different phases imposes additional barriers for the reverse martensitic transformation, and these multiple obstacles need a much larger driving force and higher activation energy.

The magnetization curves highlight the increase in the saturation magnetization of martensite as the annealing time increases, and they allow us to indicate the values and evolution of the Curie temperatures for austenite and martensite by the extrapolation of Arrott curves. The differences between them suggest increased exchange interactions in martensite induced by the thermal treatments. The thermo-magnetic measurements indicate that longer thermal treatments induce low $\gamma$ phase segregation, which depletes the austenitic matrix in 3d elements and influences magnetic ordering.

Overall, these results show that low-temperature thermal treatments can be considered an instrument for fine-tuning performance parameters and provide guidance for Ni-Mn-Ga alloy designs and processing techniques.

**Supplementary Materials:** The following supporting information can be downloaded at: https://www.mdpi.com/article/10.3390/magnetochemistry9010007/s1. Additional DSC curves at the different heating rates for the thermally treated samples (Figure S1a–c) and the $1/T$ dependence of $\log(d\alpha/dt)$ according to the Friedman model for the thermally treated samples (Figure S2a–c).

**Author Contributions:** Conceptualization, F.T. and M.S.; methodology, M.S.; software, B.P.; validation, F.T., B.P. and M.S.; formal analysis, B.P. and M.T.; investigation, F.T., M.E., B.P., C.B., M.T. and M.S.; resources, M.S. and C.B.; data curation, F.T. and M.S.; writing—original draft preparation F.T., B.P., M.T. and M.S.; writing—review and editing, F.T., B.P., C.B., M.T. and M.S.; visualization, B.P., F.T. and M.S.; supervision, M.S.; project administration, M.S.; funding acquisition, M.S. and C.B. All authors have read and agreed to the published version of the manuscript.

**Funding:** This work was supported by grants from the Romanian Ministry of Research and Innovation, CCCDI–UEFISCDI projects number PN-III-P2-2.1-PED-2019-1276, contract no. 324/2020, PN-III-P2-2.1-PED-2021-2007, contract no. 676 PED/2022 within PNCDI III, and project 35 PFE/2021.

**Institutional Review Board Statement:** Not applicable.

**Informed Consent Statement:** Not applicable.

**Data Availability Statement:** Not applicable.

**Conflicts of Interest:** The authors declare no conflict of interest. The funders had no role in the design of the study; in the collection, analyses, or interpretation of data; in the writing of the manuscript, or in the decision to publish the results.

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
