# Peer review of "Kinetics and the Effect of Thermal Treatments on the Martensitic Transformation and Magnetic Properties in Ni49Mn32Ga19 Ferromagnetic Shape Memory Ribbons"

_magnetochemistry, doi:10.3390/magnetochemistry9010007_

Round 1
Reviewer 1 Report
Minor revision
This paper discusses the magnetic properties of the NiMnGa ferromagnetic shape memory alloys. Minor issues need to revise in the paper.
Minor revision
1.Based on the Table 2 results, when the aging time is 1 hour, the transformation temperature increase. When the aging condition is 4 hours, the transformation temperature decreases. The transformation temperature increases again in 8 hours aging sample. Could the author explain the reason?
2.From the Figure 7 results, the tangent line method can determine the transformation temperature. Do these fours aging condition samples obtain the same transformation temperatures results from the previous DSC results.
Reviewer 2 Report
The article " Kinetics and the effect of the thermal treatments on the martensitic transformation and magnetic properties in the Ni49Mn32Ga19 ferromagnetic shape memory ribbons" studies the kinetics of the martensitic transformation, influence of the thermal treatments on the martensitic transformation, and related magnetic properties of the Ni49Mn32Ga19 ferromagnetic shape memory melt-spun ribbons. It also provides a method to tune the properties of Ni49Mn32Ga19 ferromagnetic shape memory ribbons.
The work is well conducted and very insightful. I have a few comments that may help the readers to get the key points more clearly:
1. In the introduction section, the authors do not explain why this composition alloy was chosen for the alloy melt-spun ribbons study?
2. The Af temperature of the alloy in Table2 is up to 351k, and the Af temperature of the alloy at 0.5T in Figure 7 is up to 370k more. Is there a better basis to explain about this difference? Or is the Arrott plot method really appropriate?
3. TEM characterization of the alloy is suggested to verify the presence of intermediate martensite from multiple angles, which is more convincing for the article.
4. Can Table1 be enriched to calculate the volume ratio and reveal the relationship between the volume ratio and the enthalpy change (or entropy change)?
5. In "Materials and Methods", EDS spectroscopy is mentioned, but the conclusion of the chemical composition of the alloy is not given in the text, instead, EDX appears in "Section 3.3". Please provide a detailed explanation.
